# A Retrospective Cross-Sectional Study Assessing Self-Reported Adverse Events following Immunization (AEFI) of the COVID-19 Vaccine in Bangladesh

**DOI:** 10.3390/vaccines9101090

**Published:** 2021-09-28

**Authors:** Arifa Sultana, Saimon Shahriar, Md. Rafat Tahsin, Sabiha Rahman Mim, Kazi Rubiya Fatema, Ananya Saha, Fahmida Yesmin, Nasiba Binte Bahar, Mithun Samodder, Md. Ariful Haque Mamun, Md. Aknur Rahman, Sonia Ferdousy, Tahmina Akter, Fahima Aktar, Md. Ruhul Kuddus, Md. Mustafizur Rahman, Md. Moklesur Rahman Sarker, Sultan Mehtap Büyüker, Jakir Ahmed Chowdhury, Abu Asad Chowdhury, Shaila Kabir, Md. Akter Hossain, Md. Shah Amran

**Affiliations:** 1Department of Pharmacy, Faculty of Pharmacy, University of Dhaka, Dhaka 1000, Bangladesh; arifa.s.meem@gmail.com (A.S.); ananyasaha029dhk@gmail.com (A.S.); fahmidayesmin189@gmail.com (F.Y.); nasibabintebahar@gmail.com (N.B.B.); 2Molecular Pharmacology and Herbal Drug Research Laboratory, Department of Pharmaceutical Chemistry, Faculty of Pharmacy, University of Dhaka, Dhaka 1000, Bangladesh; saimonshahriar19@gmail.com (S.S.); mamunphr_du@yahoo.com (M.A.H.M.); rahmanmdaknur@gmail.com (M.A.R.); fahima@du.ac.bd (F.A.); ruhulkuddus@du.ac.bd (M.R.K.); asaddg27@du.ac.bd (A.A.C.); shailakabir@du.ac.bd (S.K.); 3Department of Pharmaceutical Sciences, North South University, Plot # 15, Block # B, Bashundhara R/A, Dhaka 1229, Bangladesh; whitefang229@gmail.com (M.R.T.); kazirubiya11@gmail.com (K.R.F.); 4Department of Pharmaceutical Science, Uppsala University, 75123 Uppsala, Sweden; Sabiharahman.Mim.9635@student.uu.se; 5Department of Pharmacy, State University of Bangladesh, 77 Satmasjid Road, Dhanmondi, Dhaka 1205, Bangladesh; mithun286@gmail.com (M.S.); moklesur2002@yahoo.com (M.M.R.S.); 6Department of Pharmacy, Atish Dipankar University of Science and Technology, Uttara, Dhaka 1230, Bangladesh; soniaferdousy@adust.edu.bd; 7Department of Physiology, Dhaka Medical College, Dhaka 1000, Bangladesh; tahmina.runa.akter1@gmail.com; 8Pharmacy Discipline, Life Science School, Khulna University, Khulna 9208, Bangladesh; dipti0103@yahoo.com; 9Department of Pharmacy Services, Üsküdar University, 34662 İstanbul, Turkey; sultanmehtap.buyuker@uskudar.edu.tr; 10Department of Pharmaceutical Technology, Faculty of Pharmacy, University of Dhaka, Dhaka 1000, Bangladesh; jakir@du.ac.bd; 11Directorate General of Drug Administration, Mohakhali, Dhaka 1212, Bangladesh; akterh4@gmail.com

**Keywords:** pharmacovigilance, side-effects, SARS-CoV-2, adverse events following immunization, vaccines, cross-sectional study

## Abstract

Background: The Oxford–AstraZeneca vaccine (Covishield) was the first to be introduced in Bangladesh to fight the ongoing global COVID-19 pandemic. As this vaccine had shown some side-effects in its clinical trial, we aimed to conduct a study assessing short-term adverse events following immunization (AEFIs) in Bangladesh. Method: A cross-sectional study was conducted on social and electronic media platforms by delivering an online questionnaire among people who had taken at least one dose of the COVID-19 vaccine. The collected data were then analysed to evaluate various parameters related to the AEFIs of the respondents. Results: A total of 626 responses were collected. Of these, 623 were selected based on complete answers and used for the analysis. Most of the respondents were between 30–60 years of age, and 40.4% were female. We found that a total of 8.5% of the total respondents had been infected with the SARS-CoV-2 virus. Our survey revealed that out of 623 volunteers, 317 reported various side-effects after taking the vaccine, which is about 50.88% of the total participants. The majority of participants (37.07%, 231/623) reported swelling and pain at the injection site and fever (25.84%, 162/623); these were some of the common localized and generalized symptoms after the COVID-19 vaccine administration. Conclusion: The side-effects reported after receiving the Oxford–AstraZeneca vaccine (Covishield) are similar to those reported in clinical trials, demonstrating that the vaccines have a safe therapeutic window. Moreover, further research is needed to determine the efficacy of existing vaccines in preventing SARS-CoV-2 infections or after-infection hospitalization.

## 1. Background

COVID-19 is caused by the SARS-CoV-2-enveloped virus [1], which became a pandemic in 2020 and is still widespread [2,3]. The COVID-19 infection has the potential to cause multi-organ, systemic complications with a greater risk of thromboembolism [4,5]. As of 2 July 2021, there were more than 182 million confirmed worldwide cases of COVID-19 reported to the WHO, with more than 3 million deaths. The advent of the COVID-19 mutation and its global health implications prompted the formulation of effective and safe vaccines for emerging deadly variants [6].

A total of 2,950,104,812 vaccine doses had been given worldwide till 1 July 2021 [7]. Despite the fact that the immunization program was a long-awaited event, a large part of the global population expressed reservations about the vaccine’s efficacy and adverse effects [8]. “Short length of the vaccine development process”, “insufficient evidence to prove the efficacy of the vaccines”, “safety evaluation of vaccines in the development process”, and “potential side-effects recorded” were some of the most prevalent causes behind this [9]. Furthermore, when a global mass vaccination campaign was rolled out, certain severe and uncommon adverse events following immunization (AEFI) were observed [10].

Recent research has revealed that the Oxford-AstraZeneca’s vaccine is less effective against the aggressive B.351 strain that has recently appeared in South Africa [11]. Furthermore, researchers have discovered side effects of the Pfizer vaccine, reporting that it can induce myocarditis, arrhythmia, and even death in 1/1,000,000 individuals. This was seen in 62 men between 18–30 years aged people in Israel, two of whom died [12]. In addition, Pfizer and AstraZeneca have reported post-vaccination infection rates of 0.8 and 0.3 percent, respectively, in the UK [13]. According to a leading pandemic specialist in the United States, the Moderna vaccine caused more allergic reactions in patients who received vaccines from a certain immunization location in California. As a result, the immunization program in that centre was discontinued [14]. The effectiveness of the Sinopharm vaccine/BBIBP COVID-19 vaccine is still being assessed since its approval by the China National Medical Products Administration on 31 December 2020 [15].

Owing to these reasons, post-vaccination surveillance is crucial at this stage of vaccination rollout in Bangladesh to raise public confidence and assess the real-life effectiveness and safety of the authorized vaccines [16,17].

The outbreak occurred and spread rapidly to Bangladesh in March 2020. On 8 March 2020, the country’s Institute of Epidemiology, Disease Control and Research (IEDCR) revealed the first three known cases, and on 18 March 2020, the first death [18]. Since then, the epidemic has spread across the country, with the number of individuals infected steadily growing. Consequently, the government launched an immunization program from Dhaka on 27 January 2021, and started a nationwide COVID-19 vaccination drive on February 7. On 25 January 2021, Bangladesh received the first consignment of 5 million doses of Covishield vaccines [19]. On 2 July 2021, Bangladesh received five lakh doses of the Sinopharm vaccine from China. Beijing Bio-Institute of Biological Products Co Ltd., a Chinese National Biotech Group (CNBG) subsidiary, produces the Sinopharm vaccine [20]. Bangladesh has administered at least 2.08 million doses, according to Our World in Data’s latest update [21]. Bangladesh ranks 11th over Spain, Poland, and Canada in the worldwide vaccination race by February 20, based on the number of persons who have received nearly one dose of the COVID-19 vaccine [21]. In this survey, we aimed to conduct a cross-sectional study on the AEFIs of COVID-19 vaccines administered in Bangladesh.

## 2. Method

### 2.1. Design and Sample Selection

This survey study on AEFIs of the COVID-19 vaccine was performed online using a retrospective and cross-sectional method. We prepared a survey questionnaire after a careful review of COVID-19 data and surveillance from the Centers for Disease Control and Prevention (CDC) [22]. An extensive literature review on the associated side-effects of COVID-19 vaccines [8,9,10] and group discussion was integrated to finalize the questionnaire. Ethical approval was obtained from the human ethical review committee of the State University of Bangladesh to conduct the study (approval number: 2021-03-25/SUB/H-ERC/0005). The online questionnaire was then distributed over social and electronic media (Facebook, Twitter, WhatsApp, Email) using a snowball sampling method.

At the beginning of the COVID-19 vaccination drive, we performed pre-testing of the questionnaire by distributing it to 50 primary receivers. Different demographical parameters (i.e., age, gender, educational qualifications, residential area), pre-existing disease conditions, and side-effects associated with COVID-19 vaccine during the pre-testing are presented in Appendix A. The aim of the pre-testing was to make sure that the questionnaire was evident and unambiguous.

Later, these participants were encouraged to forward the questionnaire link to others in their social networks. The intended participants were Bangladeshi individuals not less than 18 years of age, and who could read and interpret Bangla or English. Because of the constraints of utilizing face-to-face techniques during an active outbreak, the data was solely collected using the Google Forms platform. This online form is extensively shared via social and electronic sites in Bangladesh and is widely utilized by people of all socioeconomic backgrounds and different age groups [23].

### 2.2. Questionnaire on Adverse Events following COVID-19 Vaccine Immunization 

The questionnaire was created in response to the circumstance through group discussion. The survey form consisted of seven sections containing vaccination information, health condition before and after the vaccination, associated side-effects after the vaccination, any symptom management step taken by participants, etc. The first section contained general information about this survey and asked for consent. All of the respondents were obligated to answer this section in order to continue with the survey. The second section contained personal information such as age, sex, residence, educational qualification, etc. The next section contained a question concerning the vaccination information of the current individual, including vaccine name, manufacturing company, vaccination date, dose, etc. The fourth section was specially designed for females. It contained three questions, including the pregnancy and lactation condition of the female. The following section presented several questions related to the current health status of the individuals before the vaccination. This section contained questions regarding the current COVID-19 status of the participant, preventive measures taken, such as pneumonia vaccinations or plasma therapy. This section also addressed allergic conditions, chronic diseases with current treatment patterns, previous vaccination information, etc. Most of the questions in this section were in dichotomous ‘yes’ or ‘no’ format. Section 6 in the questionnaire presented in supplementary document S1 was headlined as “After Effects Following Vaccination” and contained only two questions: (1) Had the participant been affected by COVID-19 before the first dose? And (2) did the participant face any physical discomfort? Both were in dichotomous ‘yes’ or ‘no’ format. The seventh section was only for respondents who had responded “yes” to the last question in the previous section. This section presented the type, duration, management, and treatment pattern of physical discomfort after the vaccination. The original questionnaire was prepared in English but later translated into Bangla for easy understanding.

### 2.3. Duration of the Study

The study was conducted between 1 February and 30 June 2021. A total of 21 weeks response period was allocated in order to collect replies from the COVID-19 vaccine recipients. This poll drew 626 volunteers from various socioeconomic backgrounds who had received one or two vaccination doses. In the end, 623 responses with complete answers were selected for the final analysis.

### 2.4. Statistical Analysis

The questionnaire output contained both nominal and ordinary data, and Cronbach’s alpha test was performed to predict the internal consistency of the questionnaire. The co-efficient of the reliability test conducted on the demographical and COVID-19 vaccine associated side-effect questionnaire resulted in Cronbach’s alpha value of 0.83. This value indicates a higher interrelatedness in the assessment of the questionnaire [24]. Sociodemographic parameters such as gender, age, area of residence, educational qualifications, pre-existing disease conditions, and side-effects and their severity were analysed with descriptive statistics. Cumulative and average data were presented for the assessment, and the results were presented in percentage (%).

## 3. Results

### 3.1. Demographic Analysis

This survey study focused on mass populations from different regions in Bangladesh. Of the total 623 participants, 59.6% were male, and 40.4% were female, actively answering all the constructive questions. The participants’ educational qualifications were also assessed—84.1% of them were found to have completed secondary education and higher. The participants were categorized into three different age groups: 18–30 years, 30–60 years, and above 60 years. The results from Table 1 show that the majority of vaccine recipients belong to the middle-aged group (30–60 years), corresponding to 71.4%. Participants gave responses to our survey from both rural and urban areas of the country. The respondents’ involvement was 69.02%, 29.37%, and 2.08% from urban, rural, and foreign areas (Bangladeshi diaspora), respectively.

### 3.2. Respondents Who Took Preventive Measurements

The study found that 15 out of 623 participants took pneumonia or influenza vaccines as a precaution to prevent COVID-19 infection (Figure 1). Most of them were availed from the healthcare sector as a safety measure because COVID-19 symptoms are similar to those of the flu. There were 124 participants who probably faced difficulties with this question and did not answer it.

### 3.3. Pre-Existing Diseases of the Respondents Prior to the COVID-19 Vaccination

Nearly half of the participants of this survey had chronic diseases. About 8.5% of the individuals had previously been infected with COVID-19, 13.41% had diabetes, and 17.09% had hypertension. Of the participants, 2.9% had asthma. In addition, some of the responders had listed hyperlipidaemia, renal diseases, and hypothyroidism as pre-existing conditions (Table 2).

### 3.4. COVID-19 Vaccination Side-Effects and Their Severity

Of the 623 respondents, 317 reported post-vaccination side-effects. The severity of these side-effects is classified based on their duration after the administration. Fifty-six respondents had side-effects that lasted longer than one week, reporting them as severe. There were 192 participants who reported moderate symptoms lasting longer than 1 to 3 days, and 69 had mild or less-severe symptoms (Figure 2).

The majority of participants (37.07%, 231/623) reported swelling and pain at the injection site, which is the most common localized symptom after COVID-19 vaccine administration (Table 3). There were a few generalized symptoms—fever (25.84%, 162/623) and dizziness (5.77%, 36/623)—reported by the participants. Body and joint pain (12.52%, 78/623), irritation, and burning sensation (7.70%, 48/623), classified as musculoskeletal symptoms, were reported as AEFIs. A few gastrointestinal symptoms, such as decreased appetite (1.92%, 12/623), diarrhoea (0.16%, 1/623), and nausea (2.88%, 18/623), were also reported after the vaccination. Only 0.32% of the respondents reported anaphylaxis right after the vaccination, but recovered within a few hours.

## 4. Discussion

In this survey study, we aimed to analyse COVID-19 vaccination safety. The population in Bangladesh primarily received the Covishield vaccine from Oxford-AstraZeneca (AZD1222) as of 25 June 2021. A massive part of the population was nervous and hesitant to take the vaccine due to uncertain side-effects. Our study focused on summarizing all the reported side-effects and revealing it to the larger society to clear their hesitancy. Based on the demographic data of this study, it was revealed that the educated respondents were more aware of the deadly disease, and the majority of vaccine recipients had more than a higher secondary degree. In our study, 69.02% of vaccine recipients live in urban areas, whereas less than half (29.37%) live in rural areas. Hence, there was an observable discrimination in awareness among those from rural areas. The middle and older age group—30–60 years and above 60 years—represented 71.4% and 12.51%, respectively. Almost half of the participants were found to have several pre-existing conditions. Our study shows that diabetes and hypertension were the dominant ones, representing 13.41% and 17.09%, respectively. Fifty-three participants had previously been infected by COVID-19, but still took the vaccine. Participants with high-risk conditions e.g., allergy, asthma, heart disease, kidney disease, hyperlipidaemia, and hyperthyroidism, were also included in our study. Of our respondents, 379 had no pre-existing conditions (>half of our study population) and took vaccines to avoid the severe consequences of the pandemic. COVID-19 vaccine recipients are more likely to experience some local side-effects [25,26]. A few studies from USA and China have revealed that taking the pneumonia or influenza vaccine has strong relevance with willingness to take the COVID-19 vaccine [27,28]. We were also interested in finding out a correlation between willingness to take the influenza vaccine and the COVID-19 vaccine, and found that only 2.24% of the population had taken both vaccines. Localized symptoms such as pain and swelling at the injection site, chills, joint pain, and fatigue were frequent side-effects in general, also referred by the CDC, a COVID-19 response team [29,30]. In our study, pain at the injection site, fever, swelling, nausea, headache, and decreased appetite were common side-effects that occurred at a very early stage post-vaccination, either after the first or second dose of COVID-19 vaccine. Almost 37.07% of the participants experienced these localized symptoms. More than 25% of the respondents reported fever right after the vaccination. Analyses of other studies revealed similar side-effects among vaccinated populations [26,30]. Norway reported 23 deaths after the Oxford-AstraZeneca vaccine [31]. However, these populations had a low life expectancy due to age and other disease histories. Still, this occurrence raises safety concerns associated with the COVID-19 vaccines [26], and there are several investigations into it. Thrombocytopenia and thromboembolism are also suspected to be a vaccination side-effect; however, there is not enough valid evidence to prove it [32,33]. Thus, the investigation of side-effects among the larger population in different ethnicity is important in order to formulate a real conclusion on the efficacy and safety of the COVID-19 vaccination. In our study, we chose AEFI severity based on the duration of these side-effects. We found that 192 patients, almost one-third of the study population, faced moderate side-effects that stayed for one to three days in general. Only 54 participants had severe side-effects, which continued for more than one week, and very few of them still had ongoing physical discomfort. Most of the participants were instructed to stay at the vaccination location for longer than 15 min after the administration, as per CDC guidelines [34]. During that time, they were also counselled on how to handle mild to moderate side-effects. There were only two participants who had anaphylaxis after the vaccination, which is very low in number. The reason for this could be the absence of polyethylene glycol (PEG), which was avoided as an excipient in the Oxford-AstraZeneca vaccine [35]. Moreover, this vaccine does not contain human or animal products and is thus unlikely to cause allergic reactions [13,36]. In our study, we also found a lower proportion of any kind of allergic reaction associated with the COVID-19 vaccine. Thus, our summary of the COVID-19 vaccine-associated side-effects will be helpful in eliminating confusion regarding this new vaccine, and more people will be willing to take it in a bid to help end the pandemic.

### Limitations of the Study

Although we extensively analysed our data to draw a fair interpretation, our study had a few limitations. We performed an online survey and therefore were not able to collect massive data from face-to-face interaction. We also considered responses solely based on trust rather than verified investigations by healthcare providers. In addition, we also were not able to follow up on side-effects reported after a long post-vaccination period. Thus, it is necessary to track more reports to monitor relevant side-effects as vaccination programs continue, taking into consideration the respondents’ thoughts on the necessity of vaccines to tackle the pandemic.

## 5. Conclusions

COVID-19 continues to have devastating consequences for the planet, but we must keep trying to prevent it. This study investigated the short-term side-effects of COVID-19 vaccines. The majority of the participants complained of pain and irritation at the injection site, as well as fever, body pain, swelling, vertigo, and drowsiness. Furthermore, owing to the side-effects being only short-term, only a few people had to visit a doctor or be admitted to the hospital. Large-scale follow-up research on other available vaccines should be undertaken to assess the efficiency of the vaccines in controlling and preventing SARS-CoV-2 infection in Bangladesh by determining an optimal regimen. Moreover, longitudinal surveys and associated pharmacovigilance studies should be undertaken to look into any vaccine side-effects in the long run.

## Figures and Tables

**Figure 1 vaccines-09-01090-f001:**
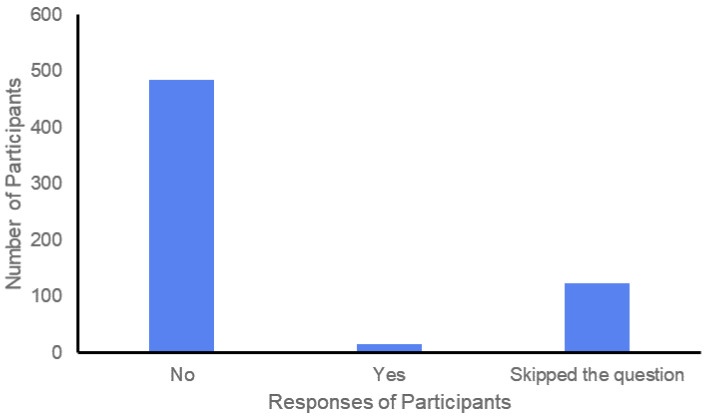
Number of participants who took pneumonia or influenza vaccine as a preventive measurement against COVID-19.

**Figure 2 vaccines-09-01090-f002:**
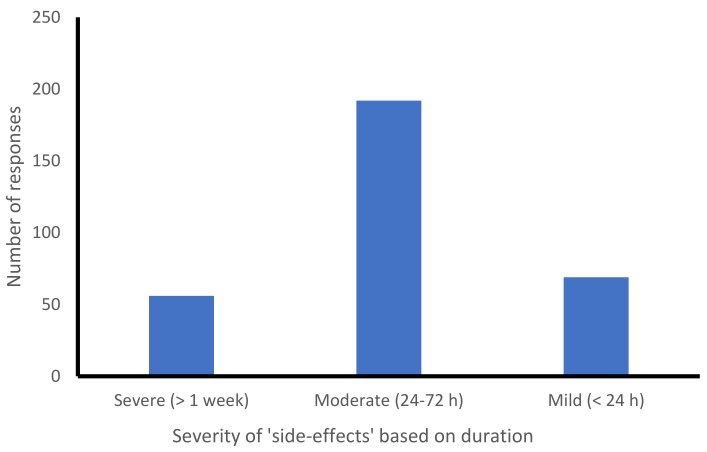
Number of participants with post-vaccination side-effects.

**Table 1 vaccines-09-01090-t001:** Demographic data of the vaccinated population from a random and cross-sectional study in Bangladesh (*n* = 623).

Variables	Outcome	Frequency	Percentage (%)
Gender	Male	372	59.6
	Female	251	40.4
Age	18–30 years	100	16.1
	30–60 years	446	71.4
	>60 years	77	12.5
Area of residence	Rural	182	29.37
	Urban	429	69.02
	Foreign	12	2.08
Educational qualification	Primary	28	4.6
	Secondary	70	11.3
	Higher secondary	53	8.6
	Undergraduate	60	9.6
	Graduate	233	37.3
	Postgraduate	179	28.6

**Table 2 vaccines-09-01090-t002:** Pre-existing disease conditions reported by the 623 respondents.

Disease Condition	Frequency	Percentage (%)
Allergy	5	0.79
COVID-19 infection	53	8.5
Asthma	15	2.39
Diabetes	84	13.41
Hypertension	107	17.09
Heart problem	23	3.67
Hyperlipidaemia	12	1.91
Hypothyroidism	4	0.63
Back and joint pain	6	0.95
Stomach upset	5	0.79
Kidney disease	4	0.63
Disease-free state	379	60.54

**Table 3 vaccines-09-01090-t003:** Side-effects after administration of the COVID-19 vaccine.

Side-Effects after the First or/and Second Dose of the COVID-19 Vaccine	Number of Symptoms	Percentage Reported (%)
Sore arm, pain and swelling at the site of injection	231	37.07
Body and joint pain	78	12.52
Fever	161	25.84
Decreased appetite	12	1.92
Nausea	18	2.88
Dizziness	36	5.77
Irritation and burning sensation	48	7.70
Diarrhoea	1	0.16
Cough, sneezing	1	0.16
Anaphylaxis	2	0.32

## Data Availability

Materials and anonymous data presented in this study are available online at https://www.mdpi.com/article/10.3390/vaccines9101090/s1.

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
