# Peer review of "A Retrospective Cross-Sectional Study Assessing Self-Reported Adverse Events following Immunization (AEFI) of the COVID-19 Vaccine in Bangladesh"

_vaccines, 2021, doi:10.3390/vaccines9101090_

Round 1
Reviewer 1 Report
This study could have been interesting, but the methodological quality, as well as important lack of information in the writing of this article make it for the moment not publishable.
The low quality of the data makes me suggest this article for a minor journal.
I have made in the PDF my comments to improve this article

Author Response
We would like to convey our warm gratitude and thankfulness to the honourable reviewers for giving time and taking the pain to assess our manuscript and come up with thoughtful and constructive comments. In the attached file below, we tried our best to reply all the comments and questions of the honourable reviewer:

Reviewer 2 Report
This work presents a study on the side effects of the COVID-19 vaccine (Oxford-AstraZeneca, Covishield.) A total of 623 samples were collected through an online survey. The results are consistent with the vaccine's official report in clinical trials. Overall, the study is timely in the prevailing circumstance. The research described in this manuscript is technically well-crafted and careful. Before I can recommend publication of this work, one concern should be addressed: Since the data was collected through an online questionnaire (Google Forms platform) and distributed widely over social media, how can you guarantee a random sampling process? The volunteers who were willing to participate in the survey might not be representative. For instance, will someone in side-effect after receiving a vaccine more likely to take the survey than someone doesn't present side effect? Elders who do not easily accessible to the social media platforms were undersampled in the survey?
Author Response
We would like to convey our warm gratitude and thankfulness to the honourable reviewers for giving time and taking the pain to assess our manuscript and come up with thoughtful and constructive comments. We have modified our original manuscript and we assured about the random sampling procedure through pretesting of the samples and distributing among different demographic characteristics.
In statistics, the term “Randomization” carries very significant meaning. In our study, we were not selecting participants randomly to compare with other groups, Therefore, the term “randomization” is not appropriate to use. So, we have replaced the word “randomization” with “Retrospective”. So, the modified title is, “A Retrospective Cross-sectional Study on Mass Population Assessing the Self-reported Adverse Events Following Immunization (AEFI) of COVID-19 Vaccine in Bangladesh”
|
We have added more detailed information as required. We have also mention it below: This survey study on AEFI of COVID-19 vaccine was performed through online using retrospective and cross-sectional method. We have prepared survey questionnaire after a careful review on COVID-19 data and surveillance from Centers for Disease Control and Prevention (CDC). An extensive literature review on associated side effects from COVID-19 vaccines and group discussion was integrated to finalize the questionnaire. An ethical approval was obtained from the human ethical review committee of the State University of Bangladesh to conduct the study. Then the online questionnaire was distributed over social and electronic media (Facebook, Twitter, WhatsApp, Email) using a snowball sampling method. At the beginning of COVID-19 vaccination, we performed the pre-testing of questionnaire by distributing among 50 primary receivers. Different demographical parameters (i.e., age, gender, educational qualifications, residential area), pre-existing disease condition, and side-effects associated with COVID-19 vaccine during pre-testing are presented in Supplementary Table I. from the table we can see that the elder group of people actively participate on the survey. In our survey based study, we showed in our result that most of our participant are educated and therefore the elder group among this survey can easily understand the questionnaire and willing to participate in the survey in their abundant time during this pandemic. In addition, the questionnaire output contained both nominal and ordinary data, and Cronbach’s alpha test was performed to predict internal consistency of the questionnaire. The co-efficient of reliability test conducted on demographical and COVID-19 vaccine associated side effect questionnaires and it results the value of 0.83 of Cronbach´s alpha. This value indicates higher interrelatedness in the assessment of the questionnaire. Sociodemographic parameters such as gender, age, area of residence, educational qualifications and the pre-existing disease conditions, side-effects and their severity were analysed with the descriptive statistics. We have presented the cumulative and average data, and expressed the results in percentage (%). |
Round 2
Reviewer 1 Report
Thank you and congratulations to the authors who worked hard on this new version of the article. My comments have been taken into account and the missing parts added.
Reviewer 2 Report
The authors have adequately answered my questions and concerns. I don't have further questions at this time.